# GENERALISED LOOKAHEAD OPTIMISER

**Costin-Andrei Oncescu & João F. Henriques**
University of Oxford
{costin,joao}@robots.ox.ac.uk

**Jack Valmadre**
The University of Adelaide
jack.valmadre@adelaide.edu.au

## ABSTRACT

The vast majority of deep learning models are trained using SGD or one of its variants. Zhang et al. (2019) suggested the Lookahead optimiser as an alternative which enjoys remarkable test performance on many established datasets and models. In this work we investigate a generalisation of this optimisation method. We validate the method empirically, generally demonstrating better results and faster convergence relative to the baselines of SGD and Lookahead.

## 1  INTRODUCTION

The Lookahead optimiser (Zhang et al., 2019) is described in Alg. 1. It comprises an outer loop, which generates a sequence of "slow weights" $\phi_t$, and an inner loop, which generates a sequence of "fast weights" $\boldsymbol{\theta}_{t,i}$. The inner loop takes $k$ steps using a gradient optimiser (e.g. Momentum SGD, Adam), and then updates the slow weights by moving in the direction of the final fast weights, scaled by a factor of $\alpha \in (0,1)$. Here, $\alpha$ and $k$ are hyperparameters that must be specified, and $\mathcal{L}^S$ represents the batch loss.

---

**Alg. 1** Lookahead optimiser

**for** $t = 1, 2, \ldots$ **do**
  $\boldsymbol{\theta}_{t,0} \leftarrow \phi_{t-1}$
  **for** $i = 1, 2, \ldots, k$ **do**
    $S \leftarrow$ sample next batch
    $\boldsymbol{\theta}_{t,i} \leftarrow \text{MomSGD}(\mathcal{L}^S, \boldsymbol{\theta}_{t,i-1})$
  **end for**
  $\phi_t \leftarrow \phi_{t-1} + \alpha(\boldsymbol{\theta}_{t,k} - \phi_{t-1})$
**end for**
**return** $\phi$

---

## 2  OUR METHOD: THE GENERALISED LOOKAHEAD OPTIMISER

Lookahead works by taking the last point in SGD's trajectory after $k$ steps, thus ignoring the $k-1$ intermediate points $\{\boldsymbol{\theta}_{t,i}\}_{i=1}^{k-1}$ (7$^{\text{th}}$ line, Alg. 1). In contrast, we aim to use all $k$ iterates to better inform our slow step. Since SGD takes steps in the direction of stochastic gradient estimates, there is a chance that the stochasticity will cause the direction to oscillate. To address this, we still seek to optimise the loss on the new batch, but constrain our search to a recently explored region (in weight space) $T$ so as to limit oscillations while preserving progress. Due to the constrained search space, this point ($\phi_t$) will achieve worse loss on the *current* batch $S'$ than an unconstrained SGD step, but still equal or better than Lookahead's $\boldsymbol{\theta}_{t,k-1}$. We choose $T$ within the affine

---

**Alg. 2** Generalised Lookahead optimiser

**for** $t = 1, 2, \ldots$ **do**
  $\boldsymbol{\theta}_{t,0} \leftarrow \phi_{t-1}$
  **for** $i = 1, 2, \ldots, k$ **do**
    $S \leftarrow$ sample next batch
    $\boldsymbol{\theta}_{t,i} \leftarrow \text{MomSGD}(\mathcal{L}^S, \boldsymbol{\theta}_{t,i-1})$
  **end for**
  $S' \leftarrow$ sample next batch
  $T \leftarrow \text{BuildRegion}(\boldsymbol{\theta}_{t,0}, \ldots, \boldsymbol{\theta}_{t,k})$
  $g \leftarrow \text{QuadFit}(\mathcal{L}^{S'}, T)$
  $\phi_t \leftarrow \arg\min_{\boldsymbol{x} \in T} g(\boldsymbol{x})$
**end for**
**return** $\phi$

---

hull of the fast weights to ensure that both a standard Lookahead jump as well as staying still (by $\phi_t \leftarrow \boldsymbol{\theta}_{t,k-1}$) are feasible. Thus, we are guaranteed to generalise both Lookahead and SGD (dropping every $(k+1)^{\text{th}}$ batch). Further parallels to existing literature are discussed in Appendix A.

The pseudocode for our method, which we refer to as Generalised Lookahead, is described in its most general form in Alg. 2. It assumes access to the objective function $\mathcal{L}$, initial parameters $\phi_0$, the number of fast updates $k$, a method to sample mini-batches, and the two additional methods:

- BuildRegion: an algorithm that, given access to the $k$ fast-weight updates, returns an affine-hull search region in which to find an update for the slow weights.

Table 1: Validation accuracy mean and standard deviation over 3 runs for the considered settings.

| Optimizer | ResNet-18 on CIFAR-10 | ResNet-18 on CIFAR-100 | LeNet-5 on CIFAR-10 | LeNet-5 on SVHN | ResNet-18 on SVHN |
|---|---|---|---|---|---|
| SGD | $82.02 \pm 0.25$ | $50.27 \pm 0.48$ | $77.46 \pm 0.53$ | $93.53 \pm 0.17$ | $95.95 \pm 0.05$ |
| Lookahead | $82.89 \pm 0.45$ | $51.08 \pm 0.20$ | $77.43 \pm 0.18$ | $93.78 \pm 0.13$ | $95.86 \pm 0.05$ |
| p=k, Hessian | $83.31 \pm 0.30$ | $55.58 \pm 0.10$ | $77.53 \pm 0.41$ | $93.61 \pm 0.08$ | $96.01 \pm 0.06$ |
| p=2, sampling | $84.04 \pm 0.39$ | | | | |
| p=2, Hessian | $83.19 \pm 0.41$ | | | | |

- QuadFit: an algorithm that, given a search region $T$ and a function $f : T \rightarrow \mathbb{R}$, returns a quadratic approximation of $f$ on $T$.

In all experiments, we set the inner optimiser to Momentum SGD as it performed best in Zhang et al. (2019), and used SciPy's Sequential Least Squares Programming (SLSQP) optimiser (Kraft et al., 1988) for the last step, although any similar solver would be sufficient.

BuildRegion yields a search space contained within the affine hull of $\boldsymbol{\theta}_{t,0}, \ldots, \boldsymbol{\theta}_{t,k}$. Even though $k$ may be a small constant, the computational cost of a quadratic approximation (via QuadFit) is a function of $\dim(T)$. Hence, we also tested PCA to lower the dimension of the affine space to a target dimension $p$. Having fixed an affine space, we let $T$ be the scaled convex hull of the projections of the fast-weight updates onto that space. Importantly, the scaling is done around the final fast-weight iterate $\boldsymbol{\theta}_{t,k}$ to allow for preserving progress. For QuadFit, we tried both sampling $\Omega(p^2)$ points within $T$ and fitting a quadratic, and using a Hessian approximation (by double back-propagation).

## 3 EXPERIMENTS AND RESULTS

We performed classification experiments on two datasets, CIFAR-10/100 (Krizhevsky et al.) and SVHN (Netzer et al., 2011), considering ResNet-18 (He et al., 2016) and LeNet-5 (LeCun et al., 1989) architectures and $k$ in $\{5, 11, 17, 21\}$, training for 200 epochs. We avoided using a learning rate decay schedule due to tuning costs for ResNet-18 on CIFAR-10/100. The other experiments converged quickly, so we used the same schedule as Zhang et al. (2019). Note that the baseline results are weaker than in the original Lookahead paper because we did not use a wide ResNet-18.

The final validations for our experiments are reported in Table 1. We only tried dimensionality reduction for ResNet-18 on CIFAR-10 (using $p = 2$) since setting $k = p$ is strictly more general. We could only afford approximating by sampling and fitting when $p = 2$ because we need $\Omega(p^2)$ samples. One interesting finding is that sampling and fitting yielded smaller approximation error than the Hessian-based approach – we credit this to the piece-wise nature of the functions, which makes local behaviour less relevant. More experiment results are available in Appendix B.

## 4 COMMENTS AND FUTURE DIRECTIONS

The method proved competitive in experiments, improving accuracy by up to 4 percent. However, it requires more tuning and was outperformed by Lookahead in one instance. Another downside is that, depending on the chosen settings, the incurred computational cost can be significant, with training taking 1.5-2.5 times longer than SGD or Lookahead. The memory consumption also grows linearly with the hyperparameter $k$. However, the heaviest computations are parallelisable; in theory, with enough parallelisation, the method could approach the speed of SGD. We treat all fast weights the same and it is conceivable that placing greater importance on later updates could help. There exist theoretical parallels to support this in the works of Zhou et al. (2021) and Scieur et al. (2018).

Finally, it remains an open question how to best employ momentum within the Generalised Lookahead framework. Throughout this project, we followed Zhang et al. (2019) in leaving the standard momentum update untouched for the slow weights. However, especially since our optimisation-based update could take a relatively large step, it is possible that a different approach to momentum (or the optimiser state in general) would improve the algorithm.

URM STATEMENT

The authors acknowledge that at least one key author of this work meets the URM criteria of ICLR 2023 Tiny Papers Track.

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

## A  DISCUSSION ON GENERALISED LOOKAHEAD

One of the most similar methods is Anderson's acceleration (Anderson, 1965) which aims to speed up fixed-point iterations by also jumping to points which are a linear combination of last $k$ steps. However, the linear combination is computed so as to minimise the residual of an approximation given by simple linear interpolation of the residuals at last $k$ steps. Scieur et al. (2016) generalised the method to an optimisation setting and no longer make the linearity assumption, ending up with an extra regularizer in the way the linear combination is chosen. This has been further improved by Scieur et al. (2018) and applied successfully in a DNN setting. Their approach becomes a particular case of ours as, together with the regularizer factor, they make a second order approximation of the loss within our search space (albeit at a much smaller computational cost) to inform the next jump. However, our approximation is widely different and approximates better by construction and the justification of the approach should better address the problem of stochasticity that is not being considered in Scieur et al. (2018).

Lastly, our approach can be seen as a heavily simplified version of Newton's method: if we let the search space be given by the affine hull of all previous steps, it will eventually stop being constraining and our quadratic approximation will be precisely that given by the second order expansion of the loss. Computing the minimum of that would then be intractable, but we get around this problem by working in a much smaller dimension. Put differently, the approach can also be considered a Quasi-Newton method as we use a low dimensional approximation of the Hessian (and gradient) to make the problem tractable.

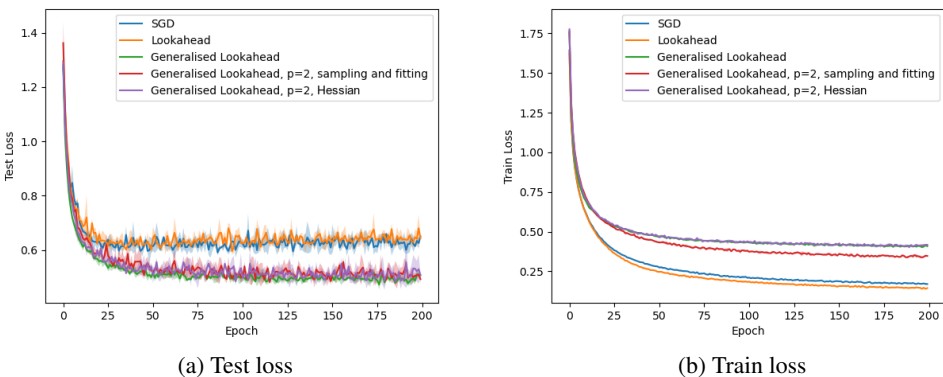

(a) Test loss          (b) Train loss

Figure 1: Performance comparison of discussed optimisers for ResNet-18 on CIFAR-10

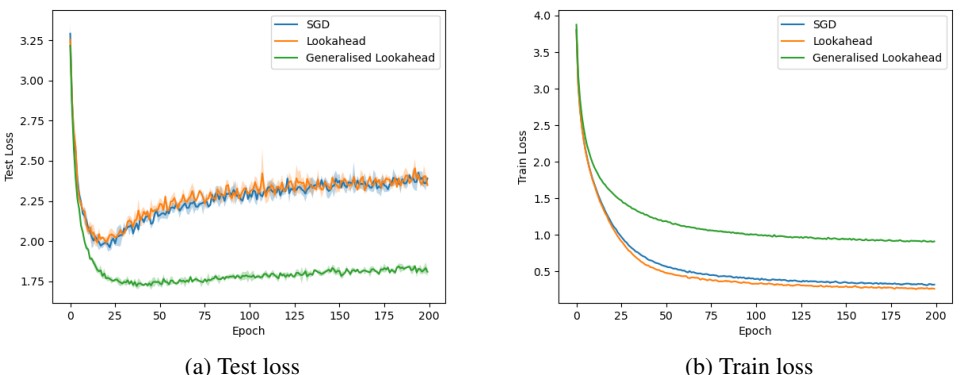

(a) Test loss          (b) Train loss

Figure 2: Performance comparison of discussed optimisers for ResNet-18 on CIFAR-100

## B EXPERIMENTS

In this section we provide more experimental results. All shaded graphs incorporate the mean and standard deviation of 3 different runs with different random seeds.

### B.1 RESNET-18 ON CIFAR-10/100

We have made the most exhaustive experiments when training a ResNet-18 on CIFAR-10/100. In the case of CIFAR-10 we also considered the option of setting $p = 2$, where we had either the Hessian-based approach or the sampling and fitting one. In the sampling and fitting case we sampled 17 points because the error was good and there was not much computational overhead. The test and train losses for CIFAR-10 and CIFAR-100 are summarised in Figure 1 and Figure 2, respectively. Generalised Lookahead outperformed by a clear margin both SGD and Lookahead and it did so by having a twice larger train loss and generalising better. In the case of CIFAR-100, the testing loss starts going up although the train loss keeps decreasing, which may indicate some overfitting but our method clearly outperforms it even before that point.

It is worth noting that the best performing setting in terms of validation accuracy was not when $p = k$, but rather when $p = 2$, in the case of approximating by sampling and fitting. This would make sense if the Hessian-based approach provides a worse approximation. To confirm this, we measured the average absolute errors for both approximations. To do so we set $p = 2$ and sampled uniformly 30 new points within $T$. In order to not bias the experiment towards one or the other, we have made the measurements while taking standard SGD steps, every $k = 17$ steps as we would in Generalised Lookahead. Note that not only are we using fresh samples to evaluate the approximation

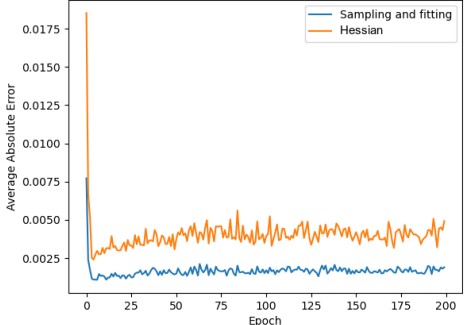

Figure 3: Average absolute error per epoch of the two considered quadratic approximation schemes

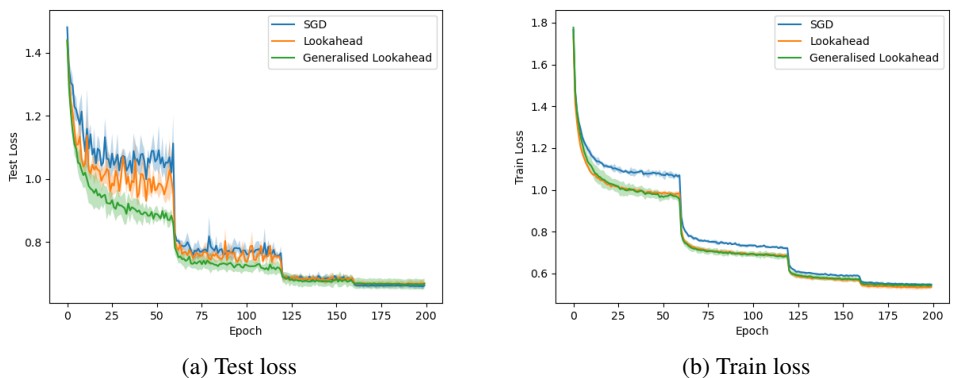

(a) Test loss                                    (b) Train loss

Figure 4: Performance comparison of discussed optimisers for LeNet-5 on CIFAR-10

error, but when doing the fitting we use least squares, so we do not aim at minimising the absolute error as measured, which means that the sampling and fitting approach does not have an unfair advantage from this point of view. The results are available in Figure 3 and confirm our theory that the sampling and fitting provides better approximation which can account for the improved performance.

### B.2    LeNet-5 on CIFAR-10

As training a LeNet-5 on CIFAR-10 converged rapidly, we used same learning rate decay scheme as Zhang et al. (2019). The test and train losses are plotted in Figure 4. As opposed to the case of ResNet-18 on CIFAR-10/100, Generalised Lookahead now has better train loss as well. This time the edge that Generalised Lookahead gives over Lookahead and SGD is much smaller in terms of final test accuracy, but it still consistently performs better than SGD, and slightly better than Lookahead. More importantly though, the speed of convergence can be seen to be higher, in the first two stages of learning rate decay having Generalised Lookahead clearly outperform SGD and Lookahead in terms of optimisation as well as generalisation.

### B.3    SVHN

Training either of ResNet-18 or LeNet-5 on SVHN converged even faster than LeNet-5 on CIFAR-10, so we kept the learning rate decay scheme in place. The train and test losses can be seen in Figure 6 for LeNet-5 and Figure 5 for ResNet-18.

In the case of ResNet-18, Generalised Lookahead still performs best, but by a small margin. However, it is worth mentioning that it is faster to optimise in the first two stages by a considerable margin. The corresponding test error is much less stable and even goes up in the second stage. We

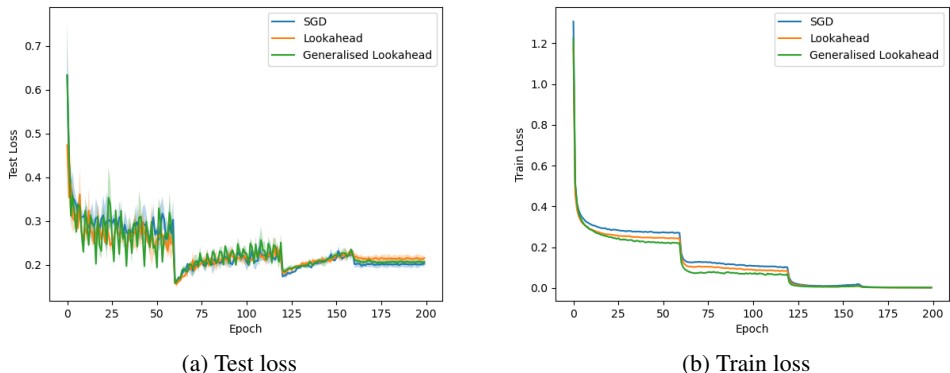

(a) Test loss

(b) Train loss

Figure 5: Performance comparison of discussed optimisers for ResNet-18 on SVHN

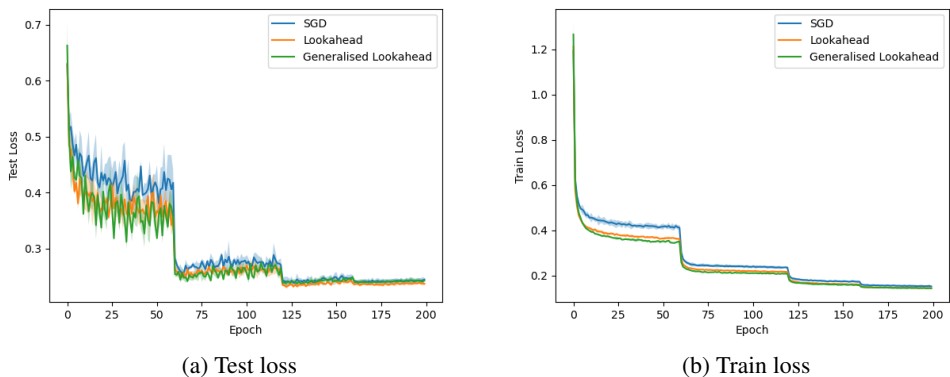

(a) Test loss

(b) Train loss

Figure 6: Performance comparison of discussed optimisers for LeNet-5 on SVHN

credit this to ResNet-18 being extremely large for how easy the task of classifying digits is, which in turn leads to overfitting.

When training LeNet-5, Generalised Lookahead still yields better train loss than both SGD and Lookahead throughout the whole training procedure, but this time it is completely outperformed by Lookahead in terms of validation accuracy. During the first two stages, it also generalises better than Lookahead and SGD but with the subsequent learning rate decays it gets worse.

