# OpenReview forum: "Generalised Lookahead Optimiser"
_ICLR.cc/2023/TinyPapers — Submitted to Tiny Papers @ ICLR 2023_

### Official Review · Reviewer_HkkN · 2023-03-27

**Confidence:** 5

**Summary Of Contributions:**

The authors propose an extension to the lookahead optimizer called generalized lookahead. The key idea is to utilize all parameters computed during the fast training process (instead of selecting the last one) to update the slow weight. The generalized lookahead achieves this by constructing an affine hull and performing quadratic approximation.

**Rating:**

Great Start (GS): a submission which meets some of the reviewing criteria but has room for improvement

**Strengths And Weaknesses:**

*Strengths*
- The findings are clearly communicated, and the paper includes an appropriate discussion of other relevant literature.
- The paper is reproducible.
- The ideas presented in this paper are interesting.

*Weaknesses*
- Some justifications in the paper are not well justified, which I mention below. However, for TinyPapers, I don't think this is a significant weakness.
- The empirical analysis is extremely limited, as the authors use a different training recipe for image classification. Hence, the numbers reported in the table are quite low than the ones presented in the literature. I don't understand why using a learning rate decay on ResNet-18 results in a higher tuning cost; the authors can use the standard scheme, which is to decay at epochs 60, 120, and 180. The current results can be biased towards the proposed methods as I imagine the generalized lookahead to have less variance update, making it less susceptible to learning rate decay. I strongly encourage the authors to evaluate the algorithm on a competitive baseline.

**Suggested Changes:**

*Suggestions*
- As mentioned above, I recommend the authors use the competitive baseline.
- The justification for why the Generalised Lookahead optimizer is weak; I recommend looking into the Noisy Quadratic Model (NQM) derivation and investigating whether the proposed method can reduce the variance.
- Authors can try gradient descent on $\alpha$s to find the best linear combination of the slow weights. Authors can use the fact that it is convex and deterministic (given a fixed batch).

*Minor*
- The sequence formula after "Lookahead works by resetting some of the progress every k steps without considering the intermediate path" should have an index $i=m$ and $m + k - 1$.
- It would help clarify if the authors describe what an affine-hull search region is (although it can be straightforward for people familiar with lookahead).

---

> ### Author Response · Authors · 2023-05-31
> **Some clarifications**
>
> Thank you very much for the feedback!
>
> The numbers in the table are actually weaker because we used ResNet-18 whereas the Lookahead paper uses a wide ResNet-18 that it confusingly refers to as plain ResNet-18. Running was too expensive and that's why we opted for the simple version.
>
> Regarding your last suggestion point, the function we optimize is a quadratic, which is convex if and only if its hessian is positive semidefinite, but we deal with the general case which may have negative curvature. The fact that it may not be convex (and indeed in practice we often had negative eigenvalues) is precisely the reason for using SLSQP (it is also a particularly low-dimensional problem since we work directly into the subspace spanned by T).
>
> Regarding the first minor point, I think what we have is correct: do note the difference between $\phi$ (the slow weights) and $\theta$ (the fast iterates - these are always reset to start off from the last slow iterate by definition, following the original Lookahead's notation).

---

### Meta-Review · Area_Chair_CVJK · 2023-04-06

**Recommendation:** Invite to archive
**Confidence:** 5

**Metareview:**

This work extends the lookahead optimizer to enhance the generalization capacity. Also, it provides some analysis of the proposed scheme.


Though this work is clear and well-written, the idea is not novel. Also, the experiments part lacks of comparisons. Furthermore, the motivation is not convincing and the contritions should be justified.

**Summary:**

This work extends the lookahead optimizer for better generalization ability of DNN.

**Comments And Feedback To The Authors:**

Please refers to the meta review part.

**Reason For Not Giving A Higher Recommendation:**

N/A

**Reason For Not Giving A Lower Recommendation:**

N/A

---

### Decision · Program_Chairs · 2023-04-07

**Decision:**

Invite to archive

**Comment:**

Novelty is expressly not a reviewing criteria, so is not considered in this decision. However, given the areas identified for improvement by the reviewers, the authors are invited to make improvements before archiving.

---

> ### Author Response · Authors · 2023-05-31
> **Opt-in for archival**
>
> We wish to opt-in for archival